# Antimicrobial Lock Therapy in Clinical Practice: A Scoping Review

**DOI:** 10.3390/microorganisms13020406

**Published:** 2025-02-13

**Authors:** Aniello Alfieri, Sveva Di Franco, Maria Beatrice Passavanti, Maria Caterina Pace, Vittorio Simeon, Paolo Chiodini, Sebastiano Leone, Marco Fiore

**Affiliations:** 1Department of Women, Child and General and Specialized Surgery, University of Campania “Luigi Vanvitelli”, 80138 Naples, Italy; anielloalfieri@gmail.com (A.A.); svevadifranco@gmail.com (S.D.F.); beatrice.passavanti@libero.it (M.B.P.); caterina.pace@libero.it (M.C.P.); 2Medical Statistics Unit, Department of Public, Clinical and Preventive Medicine, University of Campania Luigi Vanvitelli, 80138 Naples, Italy; vittorio.simeon@unicampania.it (V.S.); paolo.chiodini@unicampania.it (P.C.); 3Division of Infectious Diseases, Department of Internal Medicine, San Giuseppe Moscati Hospital, Contrada Amoretta, 83100 Avellino, Italy; sebastianoleone@yahoo.it

**Keywords:** antimicrobial lock therapy, ALT, CRBSIs, central venous catheters, continuous quality improvement, nosocomial infection, scoping review

## Abstract

Antimicrobial lock therapy (ALT) prevents microbial colonization in central vein catheters and treats existing catheter-related bloodstream infections (CRBSIs); the ALT assessment involves several key considerations. First, identifying which patients are suitable candidates is crucial. Additionally, understanding the clinical contexts in which is utilised provides insight into its applications. Examining when ALT has been employed and analyzing trends in its use over time can highlight its evolving role in patient care. Equally important is understanding how ALT is administered, including the specific agents used. Lastly, determining whether there is sufficient existing literature is essential to evaluate the feasibility of conducting future systematic reviews. This study is a scoping review adhered to the PRISMA-ScR guidelines and followed a five-stage methodological framework. Of the 1024 studies identified, 336 were included in the analysis. Findings highlight the widespread use of ethanol and taurolidine for CRBSIs prevention and the concurrent use of ALT with systemic antimicrobials to treat CRBSIs without catheter removal. ALT improves clinical outcomes, including post-infection survival and catheter retention. From our analysis, we have concluded that both an umbrella review of systematic reviews and a network meta-analysis comparing lock solutions can provide clearer guidance for clinical practice.

## 1. Introduction

### 1.1. Background

Antimicrobial lock therapy (ALT) is a specialized medical technique used to maintain a high concentration of antimicrobial agents directly within central venous catheters (CVCs). The goal of ALT is to eliminate any pathogens present in the catheter lumen and prevent the formation of bacterial biofilm, which is a common source of catheter-related bloodstream infections (CRBSIs) [1]. By keeping the catheter lumen filled with an antimicrobial solution, ALT provides targeted and sustained antimicrobial activity where it is most needed.

ALT is utilized to prevent microbial colonization in CVCs and treat existing CRBSIs; these infections can develop with both peripheral intravenous catheters (PIVCs) and CVCs, though CVCs tend to carry a higher risk of CRBSIs if compared to PIVCs, and this is due to their frequent use for long-term vascular access [2].

CVCs are widely used in inpatient and outpatient settings as they provide dependable long-term venous access for delivering medications, fluids, and nutritional support. However, CVCs and these devices remain associated with an elevated risk of CRBSIs [2]. According to the Centers for Disease Control and Prevention (CDC), CRBSIs or non-CRBSIs are included in the broader class of bloodstream infections (BSIs) and the subcategories depend on the source of contamination [3].

Approximately 90 per cent of CRBSIs in the United States are linked to CVCs, although the role of peripheral PIVCs in causing BSIs is likely underappreciated [4]. CRBSIs remain a significant clinical concern due to their impact on patient morbidity and mortality, prolonged hospital stays, and the associated increase in healthcare costs. Key risk factors for CRBSIs include contamination during catheter insertion, improper handling, and exposure during the administration of medications or parenteral nutrition [5].

In situations where catheter or port removal is not a viable option—such as in patients with limited venous access or those requiring ongoing treatments in specialized healthcare settings—a combination of systemic antimicrobial therapy and ALT is more often employed. This approach addresses both the local source of infection within the catheter and the systemic symptoms of the disease by oral or intravenous antibiotic administration, thereby enhancing the likelihood of a successful outcome, and preserving the catheter for future use.

The novelty of this scoping review lies in its objective to clearly and concisely summarize the current state of the literature on antimicrobial lock therapy (ALT). By analyzing and synthesizing available studies, this review aims to identify gaps and opportunities for future systematic investigations, such as umbrella reviews, network meta-analyses, or intervention-focused systematic reviews. This approach seeks to provide a foundation for advancing clinical practice and research in ALT.

### 1.2. Antibiotic-Based Lock Solutions

Lock therapy using antibiotic-based solutions is a method commonly used to save CVCs in CRBSIs, improve patient outcomes, and prevent catheter colonization. An antibiotic-based lock solution is instilled into the catheter lumen during periods when the catheter is not in use. Multiple studies have shown that ALT is beneficial for patients with indwelling CVCs, such as those receiving intravenous chemotherapy, parenteral nutrition, or undergoing hemodialysis [6].

Typically, very high concentrations of an antibiotic, often 100–1000 times the minimum inhibitory concentration (MIC), are combined with an anticoagulant to reduce the risk of thrombosis or catheter occlusion. Findings regarding ALT efficacy are mixed, with many clinicians attempting catheter salvage using this approach. The variability in outcomes across different studies is influenced by factors such as solution composition, temperature, and exposure time.

### 1.3. Administering Lock Solutions

The administration of lock solutions necessitates meticulous planning and adherence to aseptic techniques to optimize therapeutic efficacy and maintain catheter integrity. Prior to instillation, a thorough evaluation of the catheter is essential to identify any signs of infection, structural damage, or occlusion. The lock solution must be prepared with precision, tailored to the pathogen’s susceptibility profile, and reconstituted with sterile diluents when required. Rigorous maintenance of a sterile procedural environment is critical to mitigating the risk of contamination [7].

The lock solution should be formulated at antimicrobial or antifungal concentrations significantly exceeding systemic levels to ensure effective eradication of biofilms within the catheter lumen. The instillation volume is carefully calculated to completely occupy the catheter lumen, typically ranging from 1 to 3 mL per lumen. Before introducing the lock solution, it is imperative to confirm catheter patency and the absence of thrombotic obstructions via a sterile saline flush. Residual blood within the lumen must be cleared to prevent interference with the antimicrobial activity. The lock solution is then introduced slowly, ensuring complete filling of the lumen, and the catheter is clamped to prevent reflux or backflow [8,9].

The dwell time for the lock solution, which varies depending on the specific antimicrobial agent and the nature of the infection, generally spans from 30 min to 24 h. After the designated dwell time, the lock solution must be aspirated using a sterile syringe to avoid systemic exposure to high-concentration antimicrobials. Subsequently, the lumen should be flushed with sterile saline or a heparin solution to ensure the removal of residual solution and to maintain catheter functionality [10].

### 1.4. Common Antibiotic Agents

Several antibiotics have been explored for use in lock solutions. Among the most extensively studied are beta-lactams; ampicillin and cefazolin have demonstrated high efficacy against Gram-positive infections and compatibility with heparin in various concentrations. Other beta-lactams, including cephalosporins like cefotaxime and ceftazidime, as well as extended-spectrum agents like piperacillin, piperacillin/tazobactam, ticarcillin/clavulanate, and cefotaxime, have also been evaluated for their effectiveness in managing CRBSIs. Carbapenems, in combination with heparin, have shown fewer promising results.

Aminoglycosides such as amikacin, gentamicin, and tobramycin are frequently used in ALT, often combined with additives like heparin, citrate, or tissue-type plasminogen activator (TPA). Despite their proven effectiveness in vitro against a range of pathogens, their systemic use with a high MIC in a short volume poses toxicity risks, particularly when high concentrations are flushed into circulation. The stability of aminoglycosides in combination with heparin remains uncertain; however, citrate-based solutions have shown more consistent outcomes. Gentamicin with citrate is considered a valid option for both the treatment and prevention of CRBSIs [8].

Vancomycin, a glycopeptide antibiotic, has also been effective in ALT, especially when combined with heparin, TPA, or citrate. Similarly, teicoplanin has demonstrated its efficacy as an ALT, though the results are less consistent when combined with citrate or other agents like gentamicin and ciprofloxacin. Telavancin, another glycopeptide, shows compatibility with heparin and citrate, although clinical data are limited [11].

### 1.5. Other Antibiotics

Fluoroquinolones, such as ciprofloxacin and levofloxacin, offer additional options for managing Gram-negative CRBSIs. These agents, when used in combination with heparin at lower concentrations, show good compatibility and stability. However, issues arise with incompatibility at higher concentrations of each component [8].

Tetracyclines, particularly minocycline, have been widely utilized as lock solutions due to their efficacy against bacterial biofilms and their synergism with ion chelators. Despite their advantages, tetracyclines are incompatible with heparin. Doxycycline, combined with EDTA, represents an alternative, though the limited availability of EDTA may restrict its use [12].

Other antibiotics used in ALT include tigecycline, daptomycin, and linezolid, which are particularly effective against resistant Gram-positive infections. Tigecycline has shown favorable clinical outcomes when combined with heparin or N-acetylcysteine (NAC), although the limited number of studies suggests it should be reserved for highly specific cases. Daptomycin, for instance, requires supplementation with calcium or lactated Ringer’s solution to maintain its activity, while linezolid shows good stability with citrate or heparin, albeit with limited clinical data [13].

Linezolid appears to maintain good stability when combined with citrate or heparin; however, due to the limited availability of published clinical data, its use should be limited to specific cases with restricted treatment alternatives.

Additionally, agents like colistimethate, clindamycin, macrolides, and sulfamethoxazole/trimethoprim (SMX/TMP) have been investigated. Colistimethate and SMX/TMP may be promising options for treating CRBSIs caused by multidrug-resistant organisms (MDROs), but stability data are still insufficient [14].

### 1.6. Antifungal Agents

The use of antifungal agents in ALT is limited primarily due to insufficient compatibility and stability data, which restricts their application to select clinical cases. However, recent studies have begun exploring potential antifungal lock solutions, such as amphotericin B and echinocandins, for use in catheter-related fungal infections. Preliminary data suggest that while amphotericin B has shown some efficacy in reducing fungal colonization, stability issues remain a significant challenge. Echinocandins, such as caspofungin, are being studied for their potential to overcome biofilm-related resistance, but more research is needed to establish their safety and effectiveness in lock therapy settings [15,16].

### 1.7. Alternative Lock Solutions

Especially in the case of patients with special needs for CRBSI prevention, alternative antibiotic-free lock solutions can be considered. Ethanol-based lock therapy solutions may be utilized as an alternative option to antibiotic-based solutions in the conservative management of CRBSIs, particularly in cases where antibiotic resistance is a concern or in patients with a history of adverse reactions to antibiotics [17]. Additionally, ethanol-based solutions are often preferred in resource-limited settings due to their cost-effectiveness and ease of availability. Furthermore, a 70% ethanol lock therapy is an inexpensive and well-tolerated option for CVC salvage in patients with CRBSIs, although further studies are warranted to validate its long-term efficacy.

Ethanol is a bactericidal and fungicidal antiseptic with no cell toxicity, and concerns about the development of resistance or the promotion of cross-resistance to other antimicrobial classes are lacking.

Sanders et al. conducted a randomized, double-blind, controlled trial that compared ethanol lock solution with heparinized saline for the prevention of CRBSIs in adult cancer patients and found that the daily administration of ethanol locks into the lumens of CVCs effectively reduced the incidence of CRBSIs [17].

Then, Slobbe et al. and Kubiak et al. utilized a 70% ethanol-based lock solution in their studies. The first found that the reduction in the incidence of endoluminal CRBSIs using a preventive sole ethanol lock was non-significant [18]. The second found that the use of 70% ethanol lock therapy for CRBSIs appeared to be well tolerated and useful in a cohort comprised predominantly of cancer patients requiring long-term indwelling CVCs for chemotherapy and supportive care [19]. In a few cases, patients had persistent or recurrent bacteremia that led to CVC removal after ethanol lock therapy. However, further investigations are needed.

A work by Alonso et al. tested, in vitro, three concentrations of ethanol (25%, 40%, and 70%) with and without heparin (60 UI) at six different time points versus a 24 h preformed biofilm. They measured the reduction in the metabolic activity of the biofilm with the 40% ethanol +60 IU heparin solution administered for 72 h [20]. This solution was sufficient to eradicate the metabolic activity of bacterial and fungal biofilms, although future studies are required.

However, the use of heparin-based solutions remains frequent, which seem to be related to the development of an intraluminal biofilm [21].

Taurolidine, a derivative of the amino acid taurine, is an antimicrobial agent with broad-spectrum activity against both Gram-positive and Gram-negative bacteria. Taurolidine’s chemical mechanism, which involves reactivity with key bacterial components, minimizes the likelihood of bacterial resistance development in contrast to conventional antibiotics. Its antimicrobial activity includes methicillin-resistant Staphylococcus aureus (MRSA), coagulase-negative staphylococci (CoNS), and vancomycin-resistant enterococci (VRE). It also exhibits antifungal properties. Taurolidine has minimal reported side effects, and its use does not contribute to the development of bacterial resistance [22].

Both retrospective and prospective studies have demonstrated a reduction in CRBSIs following the use of a taurolidine lock in catheter lumens.

### 1.8. Patients’ Settings

ALT is frequently utilized in clinical settings where preserving an implanted catheter is of paramount importance to avoid more invasive and risky interventions. This strategy has shown substantial effectiveness in reducing the incidence of CRBSIs, which directly contributes to lower patient mortality rates, decreased healthcare expenses, and shorter hospital stays. Such advantages are particularly significant for patients requiring long-term catheter access, such as those undergoing chemotherapy, parenteral nutrition, or hemodialysis [22].

In intensive care units (ICUs) or pediatric wards, ALT is usually the preferred choice for patients that often show compromised venous heritage, where catheter removal and replacement would pose considerable challenges and risks [23]. In outpatient settings, ALT allows for a better continuity of care, enabling patients with chronic conditions to maintain catheter function for extended periods while minimizing the risk of infection [24].

The application of ALT in specialized settings, such as oncology wards and dialysis units, has further demonstrated its value. In these environments, where patients are already vulnerable due to immunosuppression or renal insufficiency, ALT serves as a proactive measure to prevent CRBSIs that could complicate patients’ management and lead to severe complications [23,24]. Thus, ALT is not only a therapeutic option but also a preventive tool that can enhance patient safety and quality of life.

### 1.9. Alternative Therapeutic Approaches to Antimicrobial Lock Therapy

Despite its efficacy, ALT faces challenges such as the emergence of antimicrobial resistance, potential systemic toxicity, and biofilm-associated resilience of pathogens. The search for alternative therapeutic approaches has steered research toward innovative modalities such as photodynamic antimicrobial therapy (PDT) and nanotechnology-based interventions. These approaches offer promising avenues for overcoming limitations of conventional ALT by harnessing physical, chemical, and biological principles to target pathogens with precision and minimal collateral damage [25].

#### 1.9.1. Photodynamic Therapy

Photodynamic therapy utilizes photosensitizers (PS) that, upon activation by light of a specific wavelength, generate reactive oxygen species (ROS). These ROS exert potent antimicrobial effects by damaging microbial cell walls, proteins, and nucleic acids. PDT is particularly advantageous for biofilm-associated infections due to its localized action and ability to disrupt biofilms [26,27].

Photosensitizers such as methylene blue, porphyrins, and chlorins have demonstrated significant activity against bacterial and fungal biofilms. For catheter applications, immobilized PS molecules can be integrated into catheter surfaces, creating a self-sterilizing interface upon light exposure [28].

Fiber optic or LED-based systems can be inserted into catheter lumens to activate the photosensitizers. Advances in endoluminal light delivery have enhanced the precision of PDT, ensuring minimal impact on surrounding tissues [29].

Nanotechnology-based approaches leverage the unique physicochemical properties of nanoparticles to combat infections. Their small size, high surface area, and ability to be functionalized with antimicrobial agents make them ideal candidates for addressing the challenges posed by CRBSIs [30]. Nanoparticles can be conjugated with ligands or antimicrobial peptides to enhance specificity and binding to biofilm-associated pathogens. Functionalized nanoparticles exhibit increased retention within biofilm matrices, improving their therapeutic efficacy. Among nanoparticles we find:Silver Nanoparticles that exhibit broad-spectrum antimicrobial properties, destabilizing microbial membranes and interfering with cellular functions [31].Gold Nanoparticles functionalized with antimicrobial peptides or drugs to enhance targeted delivery [32].Zinc Oxide and Titanium Dioxide in use for their photocatalytic properties, generating Reactive oxygen species under UV light and augmenting antimicrobial activity [33].Polylactic acid and poly(lactic-co-glycolic acid) capable to encapsulate antibiotics, ensuring sustained release and localized action within catheter lumens [34].Liposomes and solid lipid nanoparticles that provide biocompatible platforms for delivering antimicrobial agents directly to biofilm-embedded bacteria [35].

#### 1.9.2. Advantages and Disadvantages over Traditional ALT

One of the primary advantages of alternative modalities over traditional ALT is that they do not rely on antibiotics, thereby reducing the risk of resistance development. These methods also offer broad-spectrum activity, making them effective against MDROs. Furthermore, they demonstrate enhanced biofilm disruption, which is a critical factor in addressing catheter-related bloodstream infections (CRBSIs) [25].

However, these approaches are not without their disadvantages. A significant limitation is the restricted penetration of light into deep tissues, although advancements in near-infrared (NIR) photosensitizers may help overcome this issue. Additionally, there is a need to develop biocompatible and highly potent photosensitizer (PS) molecules [25]. The availability of these technologies remains limited worldwide, and the high cost of the materials poses a further challenge. Implementation also requires specialized training, which adds another layer of complexity.

Moreover, the long-term biocompatibility and environmental safety of nanoparticles need to be thoroughly evaluated. The standardization and approval processes for nanoparticle-based therapies remain intricate and time-consuming, further delaying their integration into clinical practice.

While both modalities face these challenges, ongoing research and technological advancements are likely to translate these promising therapies from experimental settings to widespread clinical application. These factors collectively explain why articles on these specific techniques have not been included in this scoping review.

## 2. Materials and Methods

This scoping review adhered to the methodology outlined in the PRISMA extension for scoping reviews (PRISMA-ScR) [36] and followed the recommendations provided in the JBI Manual for Evidence Synthesis [37]. This methodological approach ensured a rigorous, transparent, and reproducible process throughout the review.

This scoping review is registered on the Open Science Framework (OSF) at https://osf.io/vphwm/ (accessed on 11 November 2024). Furthermore, a detailed protocol for this review has been published [38].

The scoping review was carried out in five key phases: (1) formulation of the clinical question, (2) definition of the search strategy, (3) identification of relevant studies, (4) selection of relevant studies, and (5) data synthesis and presentation of results.

### 2.1. Clinical Question

The clinical question guiding this scoping review was the following: “What is the current clinical application of ALT in the management of bloodstream infections in patients with implanted central venous catheters?” The question was formulated using the Population/Concept/Context (PCC) framework [37]:

Population: Patients with severe infections, including specific subgroups such as pediatric patients, hemodialysis patients, oncology patients, and patients requiring parenteral nutrition.

Concept: Methodologies and practices in the use of ALT.

Context: Bloodstream infections in patients with CVCs.

### 2.2. Research Strategy and Data Sources

An effective search strategy was developed by conducting an initial survey to identify relevant keywords and MeSH terms. The databases used included MEDLINE via PubMed (n = 911), EMBASE (n = 492), Web of Science (n = 314), Cochrane Central Register of Controlled Trials (n = 32), BASE (Bielefeld Academic Search Engine by Bielefeld University Library) (n = 80), Proquest (n = 226), and OpenGrey (n = 5). From these databases, a total of 2060 studies were initially identified, and 600 duplicate records were removed prior to screening. Automation tools marked 255 records as ineligible, while an additional 345 records were removed for other reasons, resulting in 1460 records being screened. Of these, 976 studies were excluded, and 484 reports were retrieved for further review. A total of 44 of these reports were systematic reviews, which were added to identify potential additional studies and subsequently classified based on their clinical settings. A total of 436 reports were assessed for eligibility, ultimately leading to the inclusion of 335 studies in the scoping review (detailed search strings are available in Appendix B). All the screening processes and report selections are described in the PRISMA-Scr flow chart presented in Figure 1.

The search strings reported in Appendix B have been screened to obtain the 335 studies included in this scoping review.

### 2.3. Citation Management

All studies identified through the search were imported into EndNote (version 21) [39], where duplicate entries were removed using both automated and manual processes. The resulting references were organized in an electronic spreadsheet (Microsoft Excel, version 2209) [40] [Microsoft Corporation, 2020], which contained details such as year of publication, authors, title, abstract, and DOI. This table served as the basis for subsequent screening and data synthesis. Retrospective studies and case reports have been included in the Appendix A to ensure transparency, as recommended in scoping review guidelines. Additionally, clinical trials and observational studies have been synthesized into tables within the main text to provide a clear and concise summary of their findings.

### 2.4. Inclusion Criteria

Eligible studies included clinical research (randomized controlled trials, cohort studies, observational studies, case reports, or case series) published in English without restrictions on the publication date. Systematic reviews were screened to identify additional relevant studies not retrieved during the initial search.

### 2.5. Title and Abstract Screening

In the initial screening phase, two authors (A.A. and S.D.F.) independently reviewed the titles and abstracts of all retrieved articles (n = 1460). Studies deemed irrelevant to the clinical question (n = 976) were excluded in accordance with the established inclusion criteria. Discrepancies between reviewers were resolved through discussion or, if necessary, by consulting a third reviewer (M.F.).

### 2.6. Quality Assessment and Data Extraction

Full-text articles of studies deemed relevant after the title and abstract screening (n = 484) were independently assessed for quality by two authors (A.A. and S.D.F.). Quality assessment was conducted using the RoB 2 tool for randomized trials and the ROBINS-I tool for observational studiesTo address potential publication biases, we employed a multi-database search strategy and included grey literature sources (e.g., OpenGrey and BASE) to minimize selection bias. Summary images of the quality assessment results have been added to the Appendix A to enhance transparency (Appendix A).These results indicate good quality for the RCTs and a lower but still acceptable average quality for other studies.

To ensure consistency and minimize bias, data extraction was also performed independently by both authors, with specific attention paid to study type, clinical context, and outcomes.

### 2.7. Data Synthesis

The data extracted from the included studies were compiled into a structured data sheet using Microsoft Excel (version 2209) [Microsoft Corporation, 2020]. The data were subsequently imported into the R environment for analysis (RStudio, version 2022) [41]. Descriptive statistics were used to summarize the findings, which were then presented using visual aids such as graphs and tables to facilitate understanding.

## 3. Results

The results presented above answer the main questions of this scoping review.

### 3.1. Which Patients Are Candidates for ALT?

Patients eligible for ALT are those for whom catheter removal is not feasible due to limited venous access or clinical necessity. These include patients undergoing long-term therapies such as chemotherapy, parenteral nutrition, or hemodialysis. Pediatric patients and those with compromised immune systems, such as oncology patients or individuals in ICUs, are also frequent candidates for ALT. ALT has been shown to be particularly effective in pediatric oncology patients with CVCs [1,42]. Hemodialysis patients, especially those with tunneled catheters, benefit significantly from ALT to prevent infections and catheter occlusion [5,43]. Additionally, ALT plays a critical role in preserving central venous access in immunosuppressed adults and in patients receiving parenteral nutrition at home [30].

### 3.2. In What Clinical Contexts Are ALT Employed?

ALT is employed in both preventive and therapeutic contexts. Preventively, ALT reduces the incidence of CRBSIs in patients at high risk, such as those on home parenteral nutrition or chronic hemodialysis [7,44]. Therapeutically, ALT is implemented when CRBSIs occur and catheter removal is not an option, often in conjunction with systemic antibiotics. In oncology settings, ALT is particularly effective for preventing infections in long-term CVCs [18,42]. Hemodialysis units widely utilize ALT to reduce infection rates in tunneled and cuffed catheters [44,45]. Additionally, ALT has been applied successfully in pediatrics and ICUs, where the risk of catheter replacement is higher [5,24].

#### Clinical Settings

The dataset provides insights into the clinical settings in which these events occur (Figure 2). Pediatrics emerges as the most prevalent setting, with 104 occurrences, underscoring its significance in this context. Hemodialysis follows with 92 instances, highlighting its relevance in managing chronic kidney-related complications. Nutrition (66 occurrences) and tunnel procedures (65 occurrences) also feature prominently, while cancer-related settings account for 42 events, reflecting the ongoing efforts to address oncological complications. Additionally, 36 case reports are documented, indicating a sustained interest in detailed clinical descriptions. This distribution underscores the range of clinical contexts involved, spanning from chronic care management to acute interventions. It is important to note that many studies were categorized under more than one setting, reflecting the overlapping nature of clinical conditions. For example, some studies addressed both pediatric and cancer patients, while others involved tunneled catheters used in hemodialysis. This categorization highlights the complexity of managing CRBSIs in diverse patient populations, often requiring a multifaceted approach. This distribution emphasizes a particular focus on vulnerable populations, such as pediatric and hemodialysis patients, who are at high risk of CRBSIs.

The pie chart in Figure 2 illustrates the different clinical settings in which ALT has been used.

### 3.3. When Has ALT Been Used, and What Are the Trends in Its Application over Time?

ALT use has evolved significantly since its initial applications in the late 20th century. The 2000s marked a turning point with increasing adoption driven by its effectiveness in preventing CRBSIs, particularly in oncology and hemodialysis populations [46]. A peak in publications occurred between 2010 and 2015, reflecting a growing clinical interest and the development of standardized protocols [8,18]. Recent trends from 2015 to 2023 indicate sustained interest in ALT, with a particular focus on pediatric populations and antimicrobial resistance management [5,24,47]. Geographic trends highlight its strong adoption in North America and Europe, where systematic protocols and evidence-based practices have been increasingly implemented [1,7].

#### 3.3.1. Temporal Trends

The temporal trends in the dataset (Figure 3) illustrate a significant evolution in event frequency from 1984 to 2024. During the initial period from 1984 to 2001, the incidence of publication on ALT topics was relatively low and stable, with sporadic occurrences of one event per year. However, from the year 2002, an upward trajectory in event frequency became evident, marked by multiple peaks over the subsequent decades. A pronounced increase has been observed between 2006 and 2014, culminating in a peak of 23 events in 2014. Despite annual variability, including a publication decrement in 2013 and 2020, the overall trend has demonstrated a sustained growth, peaking at 26 occurrences in 2023 before a slight decrease in 2024 (until October). These variations suggest influences from external or contextual factors, such as policy shifts, technological advancements, or changing research priorities. Collectively, these trends reflect a dynamic and evolving pattern in event frequency, pointing to the underlying complexities inherent in the phenomena under investigation.

Figure 1 visually depicts the trend of publications on ALT over the years, specifically from 1984 to 2024.

#### 3.3.2. Geographic Distribution

The geographic distribution (Figure 4) of the included studies reveals a strong concentration of events in the United States, accounting for 117 occurrences. Significant activity is also observed in European countries, notably Spain (28), Italy (24), France (20), and The Netherlands (19), indicating substantial regional engagement. Other notable contributions come from Turkey (12), the United Kingdom (10), China (9), and Canada (8), underscoring a diverse international presence, albeit with fewer occurrences. A balanced participation in the study of ALT is seen in Australia, Brazil, Germany, and Greece, each with 6–7 events. Overall, the data suggest that while the phenomenon has a global footprint, North America and parts of Europe dominate, possibly due to regional research priorities, resource availability, or differing levels of engagement in the field.

The global geographic representation in Figure 4 uses color coding to highlight the countries with the highest use of the lock technique for the prevention or treatment of CRBSIs.

The geographic distribution of antimicrobial lock therapy (ALT) preferences reveals distinct regional trends, reflecting local healthcare practices, resources, and resistance profiles:**North America (USA, Canada):** The use of ethanol dominates in the USA and Canada due to its cost-effectiveness and broad antimicrobial activity. Taurolidine is another frequent choice, particularly in Canada, along with combinations like EDTA, citrate, and heparin. The USA also employs a wide variety of antibiotics, such as vancomycin, gentamicin, and ceftazidime, often paired with anticoagulants.**Europe (Western and Central):** European countries show a strong preference for taurolidine, particularly in Germany, France, The Netherlands, and Switzerland, often combined with citrate and heparin. Ethanol is also commonly used in France, Spain, and Italy, while teicoplanin, vancomycin, and gentamicin are widely utilized in various combinations for ALT. Mediterranean countries like Italy and Spain incorporate antibiotics such as ciprofloxacin and amikacin into their ALT protocols.**Asia (China, India, Turkey):** Taurolidine and ethanol are notable choices in countries like China and Turkey, reflecting a mix of affordability and effectiveness. India displays a broader use of antibiotics, including vancomycin, teicoplanin, and aminoglycosides like amikacin and gentamicin, alongside ethanol-based solutions. Antifungal agents such as amphotericin-B are also noted in the region, particularly in India and China.**Australia and New Zealand:** Ethanol-based solutions are the predominant choice in Australia, with some use of taurolidine and citrate combinations. New Zealand also follows a similar pattern, with gentamicin and heparin featuring in certain protocols.**Latin America (Argentina, Brazil, Mexico):** Latin American countries exhibit diverse ALT practices, with Argentina and Brazil favoring aminoglycosides like gentamicin and amikacin, often paired with ceftazidime, ciprofloxacin, or vancomycin. Brazil also incorporates taurolidine, ethanol, and EDTA into its protocols, showcasing flexibility in resource-limited settings.**Middle East (Egypt, Iran, Saudi Arabia, Turkey):** Taurolidine combined with heparin or citrate is commonly used in this region. Countries like Iran and Saudi Arabia also employ vancomycin and cefotaxime, reflecting a focus on Gram-positive pathogens.**Africa:** Limited data are available, but taurolidine and heparin are noted in Egypt, and combinations with cefotaxime are occasionally used.

These regional differences in ALT preferences highlight the impact of local antimicrobial resistance patterns, healthcare infrastructure, and economic considerations on treatment choices. These trends emphasize the importance of tailoring ALT strategies to regional needs while considering the global challenge of antimicrobial resistance.

### 3.4. How Is ALT Administered, Including the Use of Specific Agents Such as Antibiotics or Ethanol?

ALT is administered by filling the catheter lumen with a highly concentrated antimicrobial or antiseptic solution, which remains for a designated dwell time before being aspirated or flushed. Antibiotics like vancomycin, gentamicin, and ceftazidime are frequently used, often in combination with anticoagulants such as heparin or citrate to prevent thrombosis [48,49]. Ethanol lock therapy is a notable alternative, particularly for managing MDROs or in resource-limited settings [50]. Taurolidine-based solutions are increasingly preferred for their broad antimicrobial activity and low resistance development [51]. The dwell time typically varies from a few hours to 14 days [10,52], depending on the agent and the clinical scenario. Notably, therapy durations of 48–72 h have been associated with improved clinical outcomes, including reduced infection recurrence and higher catheter salvage rates. However, studies supporting dwell times longer than 72 h lack robust comparative data [8].

#### 3.4.1. Methodologies of ALT

The dataset sheds light on the diverse methodologies used for ALT. Ethanol monotherapy is the most frequently reported approach, with 93 occurrences, followed by antibiotic monotherapy with 49 occurrences. Taurolidine and anticoagulant monotherapies are also documented, with 34 and 20 occurrences, respectively. Combination therapies are notable, with antibiotic and anticoagulant combinations reported 46 times and taurolidine with anticoagulant reported 29 times. Other approaches, such as multiple antibiotics only (35), ethanol with antibiotic (10), and ethanol with anticoagulant (8), further highlight the diversity in ALT strategies. Though less common, combinations involving multiple agents—including ethanol, taurolidine, anticoagulants, and antibiotics—are also documented, indicating the experimental and adaptive nature of ALT practices tailored to specific clinical needs.

#### 3.4.2. Antimicrobial Lock Agents

The dataset identifies a variety of ALT agents used, either alone or in combination. Ethanol is the most frequently utilized agent, appearing in 115 instances, underscoring its widespread application in ALT. Heparin follows with 76 occurrences, reflecting its common use for its anticoagulant properties. Taurolidine (64 occurrences) and vancomycin (63 occurrences) are also frequently used, highlighting their roles in infection prevention and treatment. Gentamicin (45), citrate (35), and amikacin (21) are other notable agents often used for their antimicrobial effectiveness. Agents like ceftazidime (17), ciprofloxacin (16), teicoplanin (16), and cefazolin (15) are also employed, albeit with lower frequencies. Less commonly used agents, including amphotericin-B, EDTA, daptomycin, and cefotaxime, contribute to addressing specific therapeutic contexts. This diversity in AL agents reflects a tailored selection of treatments based on the clinical condition and patient requirements.

#### 3.4.3. Monotherapy Agents

The analysis of monotherapy agents in ALT revealed that ethanol was the most frequently utilized, featuring in 93 studies. Antibiotics were employed as a monotherapy in 49 studies, taurolidine in 34 studies, and anticoagulants in 20 studies. Additionally, three studies reported the use of other agents as monotherapy, such as nitroglycerine and sodium bicarbonate.

#### 3.4.4. Most Common Agents

The dataset also highlights specific and frequently employed combinations of agents. The combination of heparin and taurolidine is the most common, with nine occurrences, followed by citrate and taurolidine (eight occurrences). Ethanol and heparin appear frequently as well, with seven occurrences, alongside the three-agent combination of citrate, heparin, and taurolidine (seven occurrences). Other notable combinations include heparin with vancomycin (six occurrences) and cefotaxime with heparin (five occurrences). Less frequent combinations, such as gentamicin with heparin (four occurrences), citrate with gentamicin (two occurrences), and the four-agent combination of amikacin, ethanol, teicoplanin, and vancomycin (two occurrences), demonstrate the varied approaches employed to manage complex clinical situations. These combinations reflect a nuanced strategy to optimize patient outcomes by leveraging both antimicrobial and anticoagulant properties.

#### 3.4.5. Most Frequent Combinations

The dataset provides insights into the frequent combinations of antimicrobial lock agents used in practice. The pairing of gentamicin and vancomycin is the most common, with 23 occurrences, suggesting complementary antimicrobial actions. Heparin combined with taurolidine is equally common (23 occurrences), indicating a strategy to balance antimicrobial and anticoagulant effects. Other frequent combinations include citrate with taurolidine (19), gentamicin with heparin (19), and heparin with vancomycin (18). These combinations illustrate common strategies for balancing antimicrobial efficacy with anticoagulation. Additional combinations, such as amikacin with vancomycin (14) and ceftazidime with vancomycin (12), further reflect the tailored approaches in various clinical scenarios, emphasizing the importance of customized treatment protocols.

### 3.5. Is There Sufficient Literature to Support Conducting a Comprehensive Systematic Review on ALT?

The existing literature robustly supports the need for a comprehensive systematic review of ALT. Multiple systematic reviews have already explored its effectiveness in preventing and treating CRBSIs across diverse clinical contexts [44,53,54]. Given the proliferation of systematic reviews, conducting an umbrella review—a meta-analysis of systematic reviews—would provide a higher level of synthesis to guide clinical practice. Such an approach could help clarify the most effective agents, dwell times, and clinical scenarios for ALT implementation. Additionally, emerging evidence highlights the importance of non-antibiotic solutions such as taurolidine and ethanol in addressing antimicrobial resistance while maintaining catheter patency [55,56].

#### 3.5.1. Observational Studies

Observational studies have primarily focused on real-world applications, providing data on long-term outcomes, patient tolerability, and practical considerations in implementing ALT protocols. These studies consistently underscore ALT’s effectiveness in reducing CRBSI incidence and maintaining catheter function in vulnerable patient populations, such as oncology and hemodialysis patients (Table 1).

For instance, Chaftari et al. [57] demonstrated the efficacy of a multi-agent approach using nitroglycerin, citrate, and ethanol to reduce infection rates in cancer patients. Tsai et al. [24] presented a comprehensive analysis of various lock solutions, including amikacin and vancomycin, in cancer settings, illustrating the flexibility of ALT protocols. Studies focusing on pediatric oncology populations, such as those by De Sio et al. [58] and Asrak et al. [59], highlighted the utility of vancomycin and meropenem in addressing CRBSIs, reflecting tailored approaches to these high-risk groups.

In hemodialysis contexts, Parienti et al. [60] demonstrated the superiority of citrate over other solutions in reducing infection rates. Complementary studies by Krishnasami [61] and Donati [62] emphasized the roles of vancomycin and taurolidine in ensuring catheter longevity and minimizing infection risks. Additionally, Moore et al. [63] highlighted the synergistic effects of combining gentamicin, citrate, and heparin in dialysis patients. For nutritional applications, studies like Raphael [64] and Davidson [65] showcased the effectiveness of ethanol-based solutions, while Lambe [66] and Ait Hammou Taleb [67] highlighted the broad-spectrum efficacy of taurolidine.

These findings collectively underscore the diversity of ALT applications and the necessity of tailoring lock solutions to the specific needs of patient populations, underlying conditions, and clinical settings.

**Table 1 microorganisms-13-00406-t001:** Observational studies included in the review.

Author	Year	Country	Setting	Lock Solution
Abbas [68]	2009	New Zealand	Hemodialysis	Heparin, Gentamicin
Agarwal [69]	2023	USA	Hemodialysis	Taurolidine, Heparin
Ait Hammou Taleb [67]	2023	France	Nutrition	Taurolidine
Aksoy [70]	2022	Turkey	Nutrition	Taurolidine, Heparin
Ardura [71]	2015	USA	Pediatrics	Ethanol
Asrak [59]	2021	Turkey	Cancer, Pediatrics	Meropenem
Beigi [72]	2010	Iran	Generic	Vancomycin
Boyer [73]	2021	France	Generic	Ethanol
Broom [74]	2008	Australia	Generic	Ethanol
Bruno [75]	2016	Italy	Generic	Ceftazidime, Gentamicin, Vancomycin, Urokinase
Bueloni [76]	2019	Brazil	Hemodialysis	Gentamicin, Heparin, Taurolidine, Heparin
Chaftari [57]	2016	USA	Cancer	Nitroglycerin, Citrate, Ethanol
Chatzinikolaou [77]	2003	USA	Cancer, Pediatrics	Minocycline, EDTA
Chhim [78]	2015	USA	Pediatrics	Ethanol
Chiba [79]	2020	Japan	Pediatrics	Ethanol
Chiou [80]	2006	Taiwan	Hemodialysis	Cefazolin, Heparin, Mupirocin
Chong [81]	2020	Singapore	Cancer, Pediatrics	Taurolidine, Citrate
Chug [82]	2023	USA	Generic	Snapicillin
Cowan [83]	1992	USA	Generic	Vancomycin, Heparin
Davidson [65]	2017	USA	Nutrition	Ethanol
De Sio [58]	2004	Italy	Cancer, Pediatrics	Vancomycin, Urokinase
Del Pozo [11]	2009	Spain	Generic	Teicoplanin, Vancomycin
Del Pozo [84]	2009	Spain	Generic	Piperacillin–Tazobactam, Levofloxacin, Vancomycin, Teicoplanin, Gentamicin, Cotrimoxazole
Diamanti [85]	2007	Italy	Generic	Amikacin, Amphotericin-B
Donati [62]	2020	Italy	Hemodialysis	Taurolidine
El-Hennawy [86]	2019	USA	Hemodialysis	Sodium Bicarbonate
Fontseré [87]	2014	Spain	Hemodialysis	Taurolidine
Fuchs [88]	2020	Israel	Pediatrics	Ethanol
Funalleras [89]	2011	Spain	Generic	Amikacin, Ceftazidime, Ciprofloxacin
Hachem [90]	2017	USA	Generic	Minocycline, EDTA, Ethanol
Hachem [91]	2018	USA	Generic	Minocycline, EDTA, Ethanol
Hirsch [92]	2023	USA	Pediatrics	Ethanol
Hirsch [93]	2024	USA	Generic	EDTA
Hu [94]	2016	USA	Pediatrics	Ethanol
Jiménez Hernández [95]	2021	Mexico	Hemodialysis	Taurolidine, Citrate, Heparin
Krishnasami [61]	2002	USA	Hemodialysis	Vancomycin
Kubiak [19]	2014	USA	Generic	Ethanol
Lafaurie [96]	2023	France	Cancer	Vancomycin
Lambe [66]	2018	France	Nutrition	Taurolidine
Lambe [97]	2016	France	Nutrition, Pediatrics	Taurolidine, Citrate
Mandolfo [98]	2020	Italy	Hemodialysis	Vancomycin, Gentamicin, Cefazolin, Ceftazidime
Mezoff [99]	2016	USA	Pediatrics	Ethanol
Moore [63]	2014	USA	Hemodialysis	Gentamicin, Citrate, Heparin
Murray [100]	2014	The Netherlands	Hemodialysis	Taurolidine, Citrate, Heparin
Niño-Serna [101]	2023	Colombia	Pediatrics	Gentamicin, Tobramycin, Amikacin, Ethanol, Vancomycin
Öncü [10]	2014	Turkey	Generic	Ethanol
Padilla-Orozco [102]	2019	Mexico	Hemodialysis	Gentamicin
Parienti [60]	2014	France	Hemodialysis	Citrate
Peterson [103]	2009	USA	Hemodialysis	Vancomycin
Piersigilli [104]	2014	Italy	Generic	Ethanol, Micafungin
Piersigilli [105]	2022	Belgium	Pediatrics	Meropenem
Pietka [106]	2019	Poland	Nutrition	Gentamicin
Poole [107]	2004	USA	Hemodialysis	Ceftazidime, Vancomycin
Puoti [108]	2023	UK	Pediatrics	Taurolidine
Raphael [64]	2016	USA	Nutrition	Ethanol
Santarpia [46]	2002	Italy	Nutrition, Hemodialysis	Teicoplanin
Saxena [109]	2005	Saudi Arabia	Generic	Cefotaxime, Heparin
Silva [110]	2012	Portugal	Hemodialysis	Citrate
Silva [111]	2013	Brazil	Generic	Cefazolin, Gentamicin
Simon [112]	2008	Germany	Pediatrics	Taurolidine, Citrate
Smego [113]	1985	USA	Generic	Heparin
Souza Dias [114]	2008	Brazil	Generic	Heparin
Tsai [24]	2015	Taiwan	Cancer	Amikacin, Amphotericin B, Ampicillin, Cefazolin, Ceftazidime, Chlorhexidine, Ciprofloxacin, Erythromycin, Gentamicin, Minocycline, Teicoplanin, Vancomycin
Vanegas Calderon [115]	2021	USA	Nutrition, Pediatrics	Ethanol
Winnett [116]	2008	UK	Hemodialysis	Citrate

#### 3.5.2. Randomized Controlled Trials

Randomized control trials (RCTs) provide high-quality evidence by directly comparing ALT interventions to standard care or alternative treatments. These trials have extensively examined the efficacy of various agents, including ethanol, taurolidine, and antibiotics, their application settings, and the specific lock solutions employed (Table 2).

For example, Al-Hwiesh et al. [118] reported the combined efficacy of vancomycin and gentamicin in hemodialysis patients. Campos et al. [43] demonstrated the utility of minocycline and EDTA in maintaining tunneled catheters. Broom et al. [48] explored ethanol and heparin combinations, showing significant reductions in infection rates for hemodialysis patients.

RCTs such as those by Decembrino et al. [130] and Lopes et al. [149] focused on the use of ethanol in pediatric and tunneled settings, further validating its role across diverse clinical scenarios. Taurolidine-based studies, including Filiopoulos et al. [135] and Solomon et al. [165], emphasized its antimicrobial spectrum and compatibility with anticoagulants, demonstrating significant reductions in CRBSI incidences.

In hemodialysis-specific contexts, the trials by Betjes et al. [123] and Moran et al. [154] investigated the effectiveness of combinations like citrate and taurolidine or gentamicin and heparin, showcasing adaptable strategies tailored to individual patient requirements. Similarly, pediatric-focused RCTs, such as those by Lopes [149] and Filippi [136], highlighted the efficacy of ethanol and fusidic acid in managing CRBSIs in vulnerable populations.

Overall, these RCTs affirm ALT’s clinical utility across various medical contexts, supporting the need for customized approaches based on patient-specific factors and clinical scenarios.

Integrating findings from both observational studies and RCTs into a systematic review or a network meta-analysis could offer a comprehensive evaluation of ALT’s clinical benefits. Such an endeavor would address key gaps in the literature, including the long-term efficacy of non-antibiotic solutions and optimal protocols for high-risk populations.

#### 3.5.3. Systematic Review by Settings

The systematic review of the dataset reveals the settings in which these studies were conducted (Figure 5). Hemodialysis is the most frequently reviewed setting, with 12 occurrences, followed closely by pediatric settings with 10 reviews. Nutrition-related studies account for six instances, while cancer and tunnel-related reviews are represented with two and one occurrences, respectively. Additionally, 17 studies are categorized as generic, indicating a broader, non-specific focus. This distribution highlights the diverse clinical contexts that are subject to systematic review, reflecting the demand for evidence-based research across a spectrum of medical scenarios, from specialized patient populations to more general medical applications.

Figure 5 visually presents the data obtained from the analysis of the clinical settings where ALT has been utilized.

#### 3.5.4. Existing Literature to Support Conducting a Comprehensive Systematic Review on ALT

Overall, the results of this scoping review highlight the increasing attention being paid to ALT, particularly in vulnerable patient groups and across a variety of clinical settings. The global distribution of studies and the increasing number of publications over time indicate the growing recognition of ALT as a valuable strategy for the prevention and management of CRBSIs. A growing body of evidence supports the utility of ALT, especially in settings involving patients at high risk of catheter-related complications. While many studies have focused on pediatric and hemodialysis populations, further research is needed to optimize the methodologies and determine the most effective combinations of agents. The findings presented here serve as a foundation for future investigations into the efficacy of ALT and its role in improving patient outcomes. It would be interesting to develop an umbrella review based on the systematic reviews identified in the research conducted on ALT for this scoping review. A systematic review with a potential meta-analysis could clarify the antibiotic or antimicrobial doses used in the various protocols for lock solutions.

## 4. Discussion

ALT represents a valuable treatment strategy for patients in the treatment of bloodstream infections or for those at risk of CRBSIs. Despite its potential, unfamiliarity with ALT often results in delays and the underutilization of the technique. To enhance clinical outcomes, clinicians must address logistical challenges proactively and consider specific questions related to their practice settings. Establishing standardized protocols and institution-specific pathways based on available evidence and local resources will be crucial for successful ALT implementation.

ALT strategies encompass various formulations, each with distinct benefits and drawbacks. **Ethanol-based locks** offer potent antimicrobial activity, particularly against biofilm-associated pathogens, and are effective in clearing CRBSIs [79]. However, their use is associated with complications such as catheter degradation, systemic toxicity, and thrombosis risk. **Heparin-based locks**, commonly used for their anticoagulant properties, reduce the risk of catheter occlusion but lack direct antimicrobial action and may inadvertently promote biofilm formation if bacteria are already present [20]. **Taurolidine-based locks** exhibit broad-spectrum antimicrobial and anti-biofilm activity, and are particularly appealing due to their low potential for resistance development; yet, their high cost and potential for local irritation limit their widespread use [165]. **Antimicrobial-based locks**, incorporating antibiotics such as vancomycin or gentamicin, are highly effective against specific pathogens but are plagued by the risks of fostering antibiotic resistance and allergic reactions. The choice of ALT strategy requires careful consideration of patient-specific factors, catheter type, and infection risks, balancing efficacy against potential complications and long-term consequences.

ALT employs various agents, each offering distinct benefits and limitations. **Aminoglycosides** (e.g., gentamicin) are highly effective against Gram-negative bacteria but carry risks of ototoxicity and nephrotoxicity [73]. **Vancomycin** and **telavancin**, targeting Gram-positive pathogens like *Staphylococcus aureus*, are potent but may contribute to resistance and renal toxicity [137]. **Fluoroquinolones** exhibit broad-spectrum activity, particularly against Gram-negative and atypical organisms, yet their overuse raises concerns of resistance and tendon damage [8]. **Tetracyclines**, including **tigecycline**, offer broad-spectrum and anti-biofilm activity but are associated with gastrointestinal side effects and the potential for resistance [13]. **Daptomycin** and **linezolid** are effective against resistant Gram-positive pathogens, including MRSA and VRE; however, daptomycin is inactivated by lung surfactants, and linezolid risks hematologic toxicity [51]. **Colistimethate** is a last-resort option for Gram-negative MDROs but can cause nephrotoxicity and neurotoxicity. **Clindamycin** and **macrolides** provide good activity against anaerobes and certain Gram-positive bacteria, though resistance and gastrointestinal intolerance are concerns [122]. **Sulfamethoxazole/trimethoprim** offers broad-spectrum utility, but hypersensitivity reactions and hematologic side effects limit its use [14]. For fungal infections, **amphotericin B** and **echinocandins** are effective against Candida species; amphotericin B is nephrotoxic, while echinocandins are expensive but well tolerated [15]. The choice of agent in ALT should be tailored to the pathogen, infection severity, patient tolerance, and resistance patterns to optimize outcomes while minimizing risks.

Two key factors must be considered in planning ALT: the number of CVC lumens and the schedule for intravenous therapies administered through the CVC, particularly continuous infusions. In the case of multi-lumen CVCs, all lumens should be filled with the locking solution. If continuous intravenous fluids are required, rotating the lumens every 12–24 h may be beneficial.

To meet the demands of clinical practice, it is advisable to provide two or three standard lock formulations for each antibiotic agent. These may include a formulation with the antibiotic alone and another for hemodialysis-dependent patients co-formulated with high-concentration heparin (5000 units/mL). Historically, heparin has been the most frequently used anticoagulant in catheter locks, but recent evidence supports the use of alternative anticoagulants, such as ion chelators, EDTA, or citrate, in specific circumstances. A meta-analysis by Zhao et al. (2014), which included 13 randomized controlled trials, found that citrate combined with antibiotics was more effective than heparin in preventing CRBSIs in hemodialysis patients, with a lower risk of bleeding [44].

The European Renal Best Practice (ERBP) group, in 2010, issued a position statement recommending 4% citrate as the preferred agent for ALT in managing hemodialysis catheter-related bloodstream infections [174]. Citrate’s calcium-chelating properties confer both antimicrobial and anticoagulant activity, reducing the bleeding risk due to its rapid metabolism in the bloodstream. This characteristic is particularly advantageous if the citrate-containing lock is inadvertently flushed into the systemic circulation. However, citrate formulations require dilution before use, and direct intravenous infusion is contraindicated. The FDA currently recommends citrate concentrations below 4% for catheter locks [175].

The clinical data on EDTA use in catheter locks are limited but promising [43]. A clinical study on the use of minocycline–EDTA lock as adjunctive therapy is currently underway [176]. Nonetheless, EDTA availability varies significantly between countries, which presents an additional logistical challenge.

The ERBP group further supports the use of 4% citrate due to its favorable benefit–risk profile compared to higher concentrations. Citrate 4% formulations are available globally; however, many are indicated solely for apheresis procedures. In the European Union, formulations like Citra-Lock™ (Dirinco AG, Wilen Bei Wollerau, Switzerland) and taurolidine–citrate 4% in combination (Taurolock™; Tauro-Implant GmbH, Winsen (Luhe), Germany) are specifically approved for use in CVCs.

One potential limitation of calcium chelators is their incompatibility with daptomycin, which requires high calcium concentrations for efficacy. To reduce the waste of antibiotic stock solutions, particularly when only small amounts are needed for formulating the lock, it is advantageous to develop formulations with standardized expiration dates and an extended stability at room temperature (e.g., trisodium citrate 40 mg/mL and gentamicin 2.5 mg/mL, stable for 122 days). Such formulations could be prepared in bulk or using intravenous doses of gentamicin.

Costly antibiotics, such as daptomycin or linezolid, are typically reserved for specific clinical scenarios in which these antimicrobials are the optimal systemic therapy. When CRBSIs are suspected, an interdisciplinary management approach should be taken to decide between catheter removal and salvage. If catheter salvage is even remotely considered, ALT with specific antibiotics should be initiated within the first 48–72 h to prevent infection-related complications and improve the likelihood of successful catheter salvage. Although the optimal dwell time for ALT is still uncertain, most clinical studies recommend a minimum of 8 h per day, with a target of 12 h per day for optimal sterilization [177].

Several in vitro models have demonstrated a reduction in bacterial colony counts with ALT; however, the impact of shorter exposure times remains unclear [178]. Ideally, the locking solution should be kept in situ whenever the CVC is not in use. The dwell time is often constrained by the need for catheter access, particularly when the CVC is used for intravenous antibiotics or other systemic therapies.

While ALT offers numerous benefits, several potential risks must be managed. As with any solution allowed to dwell in a catheter lumen, there is a risk of occlusion, which can be mitigated by incorporating an anticoagulant into the lock solution. Flushing the lock solution may also expose patients to unnecessary systemic concentrations of antibiotics and/or anticoagulants, although this risk is minimal if the lock is aspirated correctly. Nevertheless, substances with significant toxic potential—such as aminoglycosides causing ototoxicity, or high concentrations of anticoagulants (e.g., heparin 1000 units/mL or citrate 30–46.7%) leading to severe bleeding, hypocalcemia, and arrhythmias—should be avoided [179]. Furthermore, even low-level exposure to antibiotics should be minimized to reduce the risk of developing resistance.

All potential adverse events are mostly related to the accidental flush of high MIC antibiotics or anticoagulants. This potential adverse event may happen in the case of an antibiotic lock solution administered to septic patients with CRBSIs and must be carefully weighed considering the venous heritage of the patients, the general outcome, and the need for CVC salvage.

In the case of prophylaxis, no toxic lock solutions such as ethanol ones are available, and their proper use may reduce CRBSI rates, thereby decreasing the overall need for systemic antibiotic therapy [8].

## 5. Conclusions

In modern healthcare, long-term CVCs are indispensable for patients requiring continuous therapies. ALT has proven to be an effective strategy for preventing and treating CRBSIs. Agents such as teicoplanin, gentamicin, and vancomycin have demonstrated significant efficacy in this context. However, optimizing the use of ALT requires clinicians to navigate various logistical and technical challenges with foresight and precision.

Effective ALT implementation hinges on meticulous preparation and planning. Clinicians must ensure that lock solutions are formulated correctly, make informed decisions regarding the inclusion of additives such as heparin, EDTA, or citrate, and carefully determine the appropriate timing and duration for therapy. Addressing practical considerations, such as ensuring catheter accessibility and mitigating the associated risks, is also critical to achieving successful outcomes.

Ethanol lock therapy also deserves consideration as a viable alternative, particularly in cases where antibiotic resistance is a concern or where traditional agents are not suitable. Ethanol has shown promise in preventing CRBSIs and is often favored due to its broad antimicrobial properties and cost-effectiveness. However, the use of ethanol requires careful handling and patient selection, given its potential to cause catheter damage and other adverse effects if not properly managed.

The development of local protocols is essential for integrating ALT seamlessly into clinical practice. By establishing standardized, evidence-based guidelines, healthcare providers can facilitate a more consistent and effective use of ALT, ultimately enhancing patient outcomes and reducing infection rates. These protocols serve as a bridge between clinical research and routine practice, supporting healthcare teams in delivering optimal care for patients requiring long-term CVCs.

## Figures and Tables

**Figure 1 microorganisms-13-00406-f001:**
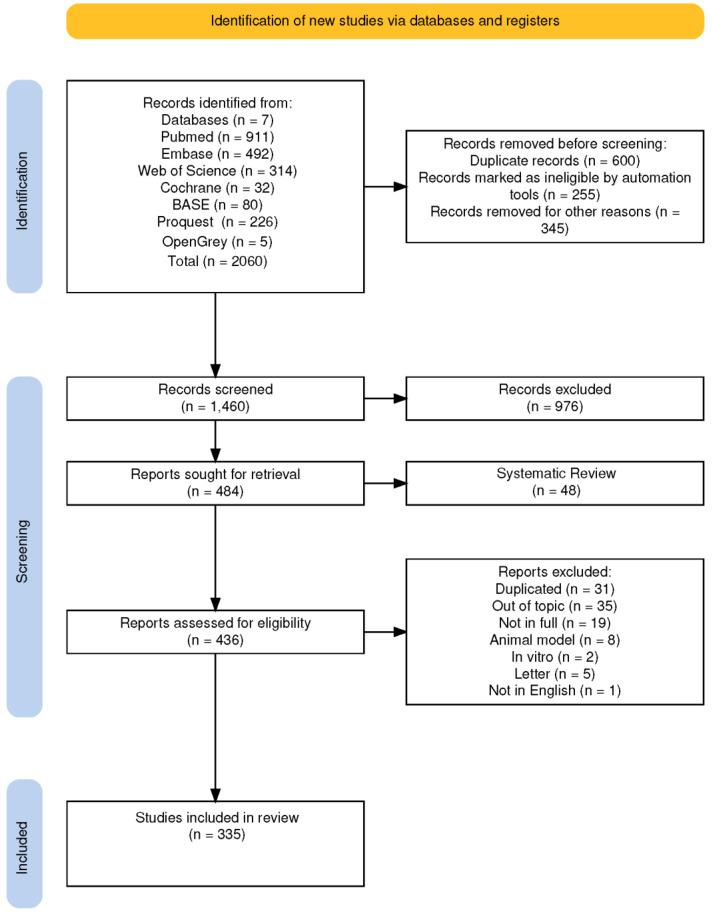
The PRISMA-ScR flow chart describes how the studies were selected.

**Figure 2 microorganisms-13-00406-f002:**
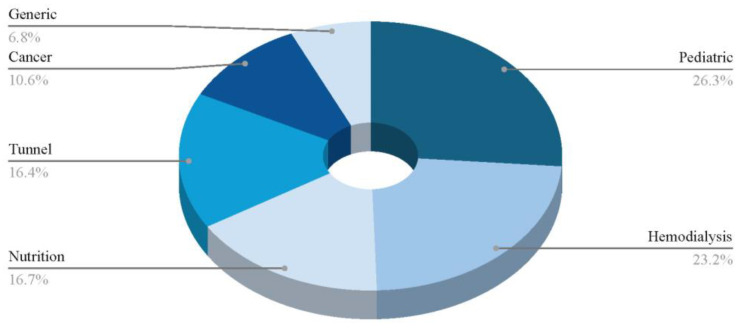
Percentages of clinical settings found in all the included studies.

**Figure 3 microorganisms-13-00406-f003:**
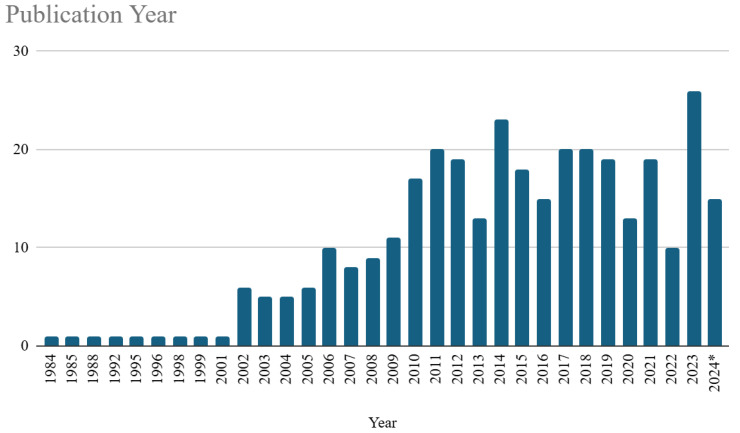
Trend of publications per year on the topic of ALT. * Until October 2024.

**Figure 4 microorganisms-13-00406-f004:**
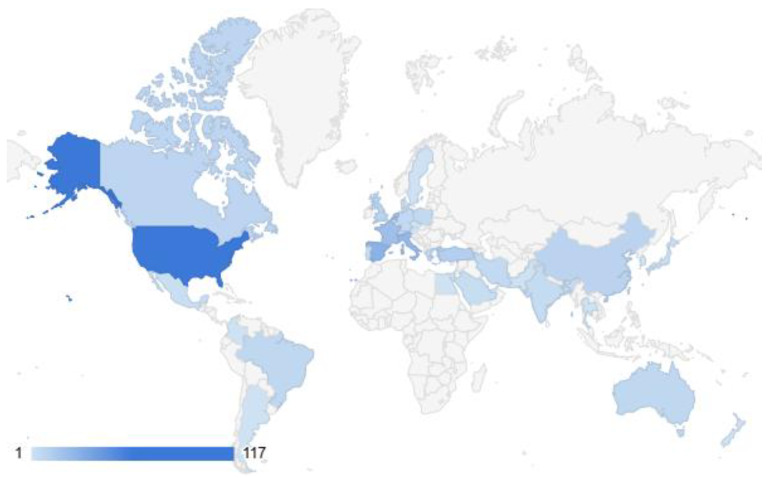
Global use of lock therapy.

**Figure 5 microorganisms-13-00406-f005:**
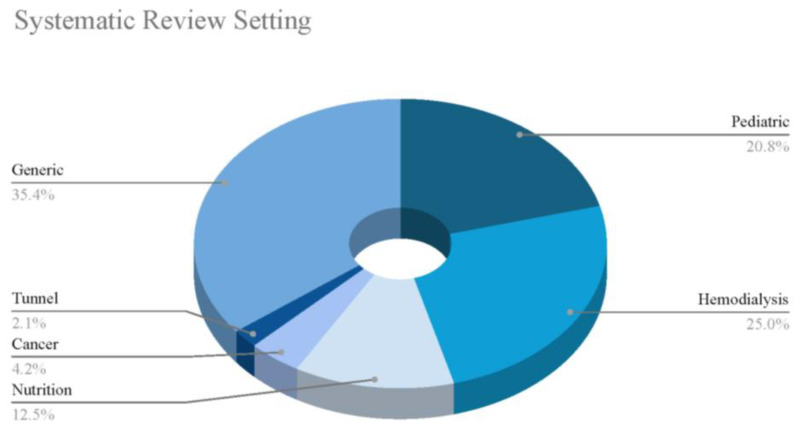
Percentage of clinical settings found in all of the systematic reviews.

**Table 2 microorganisms-13-00406-t002:** RCTs included in the review.

Author	Year	Country	Setting	Lock Solution
Al-Ali [117]	2018	Qatar	Generic	Taurolidine, Urokinase, Heparin
Al-Hwiesh [118]	2007	Saudi Arabia	Hemodialysis	Vancomycin, Gentamycin
Al-Hwiesh [119]	2008	Saudi Arabia	Hemodialysis	Vancomycin, Gentamycin
Allison [120]	2014	USA	Cancer, Pediatrics	Ethanol
Allon [121]	2003	USA	Hemodialysis	Cefotaxime, Heparin
Aniort [122]	2019	France	Tunnel, Hemodialysis	Ethanol
Betjes [123]	2004	The Netherlands	Hemodialysis	Citrate, Taurolidine
Bisseling [124]	2010	The Netherlands	Tunnel, Nutrition	Taurolidine, Heparin
Bonkain [125]	2021	Belgium	Tunnel, Hemodialysis	Taurolidine, Heparin
Broom [126]	2009	Australia	Tunnel, Hemodialysis	Ethanol
Broom [48]	2012	Australia	Tunnel, Hemodialysis	Ethanol, Heparin
Campos [43]	2011	Brazil	Tunnel, Hemodialysis	Minocycline, EDTA
Carratalà [127]	1999	Spain	Cancer	Heparin, Vancomycin
Colì [128]	2010	Italy	Tunnel, Hemodialysis	Urokinase
Davanipur [129]	2011	Iran	Hemodialysis	Vancomycin, Heparin
Decembrino [130]	2014	Italy	Tunnel	Ethanol
Dogra [131]	2002	Australia	Tunnel, Hemodialysis	Gentamicin, Citrate
Dümichen [132]	2012	Germany	Cancer	Taurolidine, Heparin
Ezzat [133]	2023	Egypt	Hemodialysis	Taurolidine, Heparin
Feeney [134]	2022	USA	Nutrition	Ethanol
Filiopoulos [135]	2011	Greece	Hemodialysis	Heparin, Gentamicin, Taurolidine, Citrate
Filippi [136]	2007	Italy	Pediatrics	Fusidic Acid, Heparin
Garland [137]	2005	USA	Pediatrics	Vancomycin, Heparin
Gudiol [138]	2020	Spain	Cancer	Taurolidine, Citrate, Heparin
Gudiol [139]	2018	Spain	Tunnel, Cancer	Taurolidine, Citrate
Handrup [140]	2013	Denmark	Tunnel, Cancer, Pediatrics	Taurolidine
Islam [141]	2024	Turkey	Tunnel	Sodium Bicarbonate, Heparin
Kanaa [142]	2015	UK	Tunnel, Hemodialysis	Heparin
Kayton [143]	2010	USA	Pediatrics	Ethanol
Khosroshahi [144]	2015	Iran	Hemodialysis	Ethanol
Kim [145]	2006	Korea	Hemodialysis	Cefazolin, Gentamicin, Heparin
Klek [146]	2015	Poland	Nutrition	Taurolidine
Lesens [147]	2024	USA	Generic	Ethanol
Longo [148]	2017	Switzerland	Cancer	Taurolidine, Citrate
Lopes [149]	2019	Brazil	Pediatrics	Ethanol
Luiz [150]	2017	Brazil	Hemodialysis	Minocycline, EDTA, Heparin, Citrate
Łyszkowska [151]	2019	Poland	Pediatrics	Taurolidine
Maki [152]	2011	USA	Generic	Citrate, Methylene Blue, Methylparaben, Propylparaben
Moghaddas [153]	2015	Iran	Tunnel, Hemodialysis	Cotrimoxazole, Heparin
Moran [154]	2012	USA	Tunnel, Hemodialysis	Gentamicin, Heparin
Mortazavi [155]	2011	Iran	Tunnel, Hemodialysis	Cefotaxime, Heparin
Nassiri [156]	2023	Iran	Generic	Taurolidine, Citrate, Vancomycin, Heparin
Nori [157]	2006	USA	Hemodialysis	Gentamicin, Minocycline
Pérez-Granda [158]	2014	Spain	Generic	Ethanol
Pérez-Granda [49]	2020	Spain	Generic	Heparin
Rijnders [159]	2019	The Netherlands	Hemodialysis	Cotrimoxazole, Ethanol, EDTA
Rijnders [160]	2005	Belgium	Tunnel	Vancomycin, Ceftazidime
Salonen [161]	2018	USA	Nutrition	Heparin, Ethanol
Schoot [162]	2015	The Netherlands	Cancer, Pediatrics	Ethanol
Slobbe [18]	2010	The Netherlands	Generic	Ethanol
Sofroniadou [163]	2012	Greece	Hemodialysis	Linezolid, Vancomycin
Sofroniadou [164]	2017	Greece	Tunnel, Hemodialysis	Ethanol, Heparin
Solomon [165]	2010	UK	Tunnel, Hemodialysis	Taurolidine, Heparin
Souweine [50]	2015	France	Hemodialysis	Ethanol
Tribler [166]	2017	Denmark	Nutrition	Taurolidine, Citrate, Heparin
van den Bosch [167]	2024	The Netherlands	Tunnel	Taurolidine, Citrate, Heparin
Wang [168]	2012	China	Hemodialysis	Ceftazidime, Heparin
Wolf [169]	2017	USA	Cancer, Pediatrics	Ethanol
Wolf [170]	2018	USA	Pediatrics	Ethanol
Worth [171]	2014	Australia	Tunnel, Cancer	Ethanol, Heparin
Wouters [172]	2018	The Netherlands	Nutrition	Taurolidine
Zhang [173]	2009	China	Tunnel, Hemodialysis	Gentamicin, Heparin

## Data Availability

The corresponding author can provide databases and literature screenings upon valid request.

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
