# Peer review of "Antimicrobial Lock Therapy in Clinical Practice: A Scoping Review"

_microorganisms, 2025, doi:10.3390/microorganisms13020406_

Round 1

Reviewer 1 Report

Comments and Suggestions for Authors

Thank you for the opportunity to review your article, which was of considerable interest to me. I believe that the topic of the article is relevant, which is associated with advances in the treatment of severe diseases requiring long-term use of central venous catheters, with corresponding infectious complications, and widespread antibiotic resistance. This topic is of particular practical importance for patients in intensive care units. The authors have done a great job of searching and selecting relevant publications for their review, using modern tools to assess their quality. I have several questions: 1) the authors selected 335 publications for the review, but only 50 references; 2) how were possible publication biases assessed; 3) it would be interesting if the authors described in more detail the methods of administering antimicrobial, antifungal drugs, or ethanol into the catheter during ALT; 4) in Figure 2, it is not entirely clear what Case Reports means and what cases these are; 5) It would be interesting to know the preferences in antimicrobial drugs for ALT in different countries

Author Response

Thank you for your thoughtful comments on our manuscript. We are pleased to address each of your points and have implemented revisions to improve the clarity and rigor of the manuscript. Below, we provide detailed responses to your questions and concerns:

  1. The authors selected 335 publications for the review, but only 50 references:

The discrepancy between the number of publications identified and the number of references included in the manuscript arises from the nature of this scoping review. While 335 studies were included in the analysis, the references cited in the manuscript represent those studies that directly informed the synthesis of results or provided essential background information. To address this, we have added a section summarizing the observational studies and RCTs, along with their respective citations, in the main text. [Lines 687-747]  Additionally, the retrospective studies and case reports have been included in the supplementary materials. This approach aligns with scoping review guidelines, which prioritize comprehensive data synthesis while maintaining a concise reference list. We will ensure this explanation is clear in the Methods section [Lines 394-398].

  1. How were possible publication biases assessed?

Following the suggestion, we have clarified in the text how publication biases were assessed and the tools used for quality evaluation, ensuring these points are explicitly addressed in the Methods section. [lines 422-427]. To further enhance transparency, summary images of the quality assessment results have been added to the supplementary materials.

  1. It would be interesting if the authors described in more detail the methods of administering antimicrobial, antifungal drugs, or ethanol into the catheter during ALT;

As suggested, we have expanded the paragraph in the introduction to provide additional detail about the technique of administration. This revised section now includes more comprehensive information to better contextualize the methodology described in the manuscript. Lines[ 114-139]

  1. Clarification on Figure 2 (Case Reports):

As suggested, we have added specific details about the case reports to the supplementary materials. Additionally, we updated Figure 2 to improve clarity by focusing solely on the clinical settings in which the studies were conducted, ensuring a more streamlined and comprehensible representation.

  1. Preferences in antimicrobial drugs for ALT in different countries:

This is an excellent suggestion, and we have now included a summary of geographical trends in the selection of antimicrobial agents for ALT.  [lines 556-597]

Additional Notes:

We have also conducted a thorough revision of the manuscript to ensure consistency and address any remaining ambiguities. Updated figures and tables reflecting these changes have been added to support the responses provided above.

We hope these revisions address your comments satisfactorily and improve the overall quality of our manuscript. Thank you again for your constructive feedback and the opportunity to refine our work.

Reviewer 2 Report

Comments and Suggestions for Authors

I have read this paper as a ‘strategic’ explorative review in preparation of or to identify topics for targeted meta-analytical research projects on a specific set of interventions to lock catheters to prevent CLABSI, or CRBSI or to co-treat in the event of infections.

I highly value your methodological approach, and respect for the PRISMA extension for scoping reviews.

How have you subdivided between 1.3 and 1.4 ?

Figure 2, typo: hemodialysis, similar for figure 5

Figure 1: please check the sequence, figure 1 follows figure 2

Related to figure 1, I assume that 2024 was ‘incomplete’, can you add something on this in the legend of figure 1.

To be open on this paper, I initially was unsure to either or not act as reviewer, because of the poorly developed abstract. After reading the full paper, there is for value in your findings, but they are at present too hidden, so that I highly recommend to come up with a better ‘explaining’ and ‘selling’ abstract. Your data and analysis are valuable, make this clearer by adding a robust abstract 

Author Response

Thank you for your valuable and insightful feedback on our manuscript. We have carefully addressed each of your comments and implemented revisions to enhance the clarity and rigor of the manuscript. Below, we provide detailed responses to your points:

  1. How have you subdivided between 1.3 and 1.4?
    We decided to divide the description of the common antimicrobials used in ALT into two paragraphs based on their frequency of administration. This classification was determined by analyzing data from the literature and the studies included in our review.
  2. Figure 2, typo: hemodialysis, similar for Figure 5
    Figure 2 represents the percentage distribution of clinical settings found in all the included studies, while Figure 5 represents the percentage of clinical settings in the included systematic reviews . We have corrected the typos in the titles and description notes of Figures 2 and 5:
  • Figure 2: Percentage of clinical settings found in all the included studies
  • Figure 5: Percentage of clinical settings found in all the systematic reviews
  1. Figure 1: Please check the sequence, figure 1 follows figure 2
    We have reviewed and adjusted the figure sequence. The figure originally labeled as Figure 3 now appears as Figure 1.
  2. Related to Figure 1, I assume that 2024 was ‘incomplete.’ Can you add something on this in the legend of Figure 1?
    We have added a footnote to the updated Figure 1 (formerly Figure 3) indicating the specific date in 2024 when the search string was last updated.
  3. Selling Abstract
    We have revised and improved the abstract to make it more comprehensive and engaging while adhering to the 200-word limit. [lines 20-52]

Revised Abstract:
Antimicrobial lock therapy (ALT) prevents microbial colonization in central venous catheters and treats catheter-related bloodstream infections (CRBSIs). We summarize the current state of the literature and provide insights on ALT by addressing five key questions:

  1. Which patients are candidates for ALT?
  2. In what clinical contexts is ALT employed?
  3. When has ALT been used, and what are the trends in its application over time?
  4. How is ALT administered, including specific agents such as antibiotics or ethanol?
  5. Is there sufficient existing literature to support conducting a comprehensive systematic review on ALT?

This scoping review adhered to the PRISMA-ScR guidelines and followed a five-stage methodological framework. Of the 1024 studies identified, 336 were included in the analysis. Findings highlight the widespread use of ethanol and taurolidine for CLABSI prevention and the concurrent use of ALT with systemic antimicrobials to treat CRBSIs without catheter removal. ALT improves clinical outcomes, including post-infection survival and catheter retention.

From our analysis, we conclude that both an umbrella review of systematic reviews and a network meta-analysis comparing lock solutions can provide clearer guidance for clinical practice.

We appreciate your constructive feedback and hope that the revisions satisfactorily address your comments. Thank you for helping us improve our manuscript.

Additional Notes:

We have also conducted a thorough revision of the manuscript to ensure consistency and address any remaining ambiguities. Updated figures and tables reflecting these changes have been added to support the responses provided above.

We hope these revisions address your comments satisfactorily and improve the overall quality of our manuscript. Thank you again for your constructive feedback and the opportunity to refine our work.

Reviewer 3 Report

Comments and Suggestions for Authors

Thank you very much for the opportunity to review the article entitled “Antimicrobial Lock Therapy in Clinical Practice: A Scoping Review"

Overall, the manuscript appears to be well-written. In addition, I think that the manuscript might deserve publication in  Microorganisms after some points are dealt with and some missing details are added prior to publication as follows:

1-    In the introduction section, the authors should discuss the novelty and motivations behind their current work.

2-    Alternative therapeutic approaches to antimicrobial lock therapy must be discussed in the manuscript.

3-    Please include in the text the benefits and drawbacks of the various Antimicrobial Lock Therapy approaches.

4-    The authors must address in the text the photodynamic and nanotechnology-based antimicrobial therapy.

5-    In addition to the References list, other recent research on the use of light and nanotechnology as alternative approaches for antimicrobial lock therapy is encouraged to be reviewed and included if beneficial:

Ø    Antibiotics (Basel). 2022 Dec 16;11(12):1826. doi: 10.3390/antibiotics11121826.

Ø    Lasers Med Sci 39, 144 (2024). https://doi.org/10.1007/s10103-024-04080-5

Author Response

Dear Reviewer,

Thank you for your thoughtful and detailed feedback on our manuscript, “Antimicrobial Lock Therapy in Clinical Practice: A Scoping Review.” We have carefully considered your suggestions and implemented the necessary revisions to address your comments. Below, we provide a point-by-point response:

  1. In the introduction section, the authors should discuss the novelty and motivations behind their current work.
    As suggested, we have revised and expanded the introduction. In this section, we emphasized the novelty of the review, focusing on its role in addressing critical gaps in the understanding and application of antimicrobial lock therapy (ALT). The additional text (lines 91-97) elaborates on how this review synthesizes current literature, identifies emerging trends and challenges, and highlights opportunities for future systematic reviews or meta-analyses. This addition provides a clear rationale for the study and strengthens its academic contribution.
  2. Alternative therapeutic approaches to antimicrobial lock therapy must be discussed in the manuscript.
    4. The authors must address in the text the photodynamic and nanotechnology-based antimicrobial therapy.
    We have added a new section, “Alternative Therapeutic Approaches to Antimicrobial Lock Therapy,” to address these comments. This section highlights innovative modalities, such as photodynamic antimicrobial therapy (PDT) and nanotechnology-based interventions, which offer promising alternatives to traditional ALT. (lines 273-343)
  3. Please include in the text the benefits and drawbacks of the various Antimicrobial Lock Therapy approaches.
    We have added detailed descriptions of the benefits and drawbacks of different ALT strategies (lines 800-841)
  4. In addition to the References list, other recent research on the use of light and nanotechnology as alternative approaches for antimicrobial lock therapy is encouraged to be reviewed and included if beneficial.
    We have added the following articles to the References list, as suggested:
  • Antibiotics (Basel). 2022 Dec 16;11(12):1826. doi: 10.3390/antibiotics11121826.
  • Lasers Med Sci 39, 144 (2024). https://doi.org/10.1007/s10103-024-04080-5.

While these articles provide valuable insights, the innovations discussed are not yet prevalent in clinical settings and therefore were not included in the scope of this review. However, we recognize their significance and intend to explore these topics further in future research.

We appreciate your constructive feedback and hope the revisions address your concerns effectively. Thank you for your time and effort in reviewing our manuscript.

Additional Notes:

We have also conducted a thorough revision of the manuscript to ensure consistency and address any remaining ambiguities. Updated figures and tables reflecting these changes have been added to support the responses provided above.

We hope these revisions address your comments satisfactorily and improve the overall quality of our manuscript. Thank you again for your constructive feedback and the opportunity to refine our work.

Round 2

Reviewer 3 Report

Comments and Suggestions for Authors

The authors have made reasonable changes to the manuscript in response to my previous suggestions and concerns. I believe the manuscript has all the information and is ready for publication as a review article in the Journal "Microorganisms ".

Author Response

  1. we rephrased the bullet points into a concise narrative structure.
  2. we stated that this study is a scoping review and updated the manuscript type as "Systematic Review" in the appropriate place.